# The Clinical Feature and Treatment Outcome of Ocular Melanoma: A 34-Year Experience in a Tertiary Referral Center

**DOI:** 10.3390/cancers13235926

**Published:** 2021-11-25

**Authors:** Yu-Yun Huang, Tzu-Yu Hou, Wei-Kuang Yu, Chieh-Chih Tsai, Shu-Ching Kao, Wen-Ming Hsu, Jui-Ling (Catherine) Liu

**Affiliations:** 1Department of Ophthalmology, Taipei Veterans General Hospital, Taipei 11217, Taiwan; yyhuang13@vghtpe.gov.tw (Y.-Y.H.); tyhou@vghks.gov.tw (T.-Y.H.); wkyu3@vghtpe.gov.tw (W.-K.Y.); sckao@vghtpe.gov.tw (S.-C.K.); wmhsu@s.tmu.edu.tw (W.-M.H.); jlliu@vghtpe.gov.tw (J.-L.L.); 2School of Medicine, National Yang Ming University, Taipei 11221, Taiwan; 3School of Medicine, National Yang Ming Chiao Tung University, Hsinchu 30010, Taiwan; 4Department of Ophthalmology, Kaohsiung Veterans General Hospital, Kaohsiung 813414, Taiwan; 5Department of Ophthalmology, Taipei Medical University-Shuang-Ho Hospital, Ministry of Health and Welfare, New Taipei City 23561, Taiwan

**Keywords:** conjunctival melanoma, eyelid melanoma, ocular melanoma, orbital melanoma, uveal melanoma

## Abstract

**Simple Summary:**

Although uncommonly encountered, ocular melanoma may threaten the vision and even the life of a patient. The clinical feature and long-term results of ocular melanoma have rarely been described in the Asian population, as it has a low incidence for non-Caucasians. Therefore, we investigated the epidemiology, clinical manifestation, treatment strategy, and long-term outcome of ocular melanoma in a tertiary referral center in Taiwan.

**Abstract:**

Malignant melanoma can arise from melanocytes in various structures of the eye, orbit, and ocular adnexa. We reviewed the clinical features and long-term results of all subjects with histologically proved melanoma originating from any of the ocular and periocular structures in a tertiary referral center. Overall, 88 patients including 47 men were recruited. The tumor was primarily located in the uvea, followed by the conjunctiva, orbit, eyelid, and lacrimal sac. Patients with uveal melanoma were diagnosed at a relatively younger age (47.0 years), while those with orbital and eyelid melanomas were older at presentation (79.5 years and 78.5 years, respectively). The overall local recurrence rate was 9% at a median follow-up of 41.0 months, among which orbital and eyelid melanomas recurred most commonly. The overall mortality rate was 41% in a median duration of 27.2 months (IQR, 13–58 months) from diagnosis, with the highest for lacrimal sac melanoma, followed by melanoma of the orbit, uveal, conjunctiva, and eyelid. Despite prompt local control, the risk for metastasis and mortality was high. Therefore, efficient modalities for early diagnosis and treatment of ocular melanoma are necessary.

## 1. Introduction

Ocular melanoma refers to malignant melanocytic proliferation involving any structure of the eye, orbit, or ocular adnexa. It is far less encountered than cutaneous melanoma. According to the analysis from the National Cancer Data Base (NCDB) in the United States on melanoma with a known primary location, only 5.5% had an ocular origin [1]. Although rare, uveal melanoma represents the most common primary intraocular malignancy in adults [2]. In fact, ocular melanoma leads to substantial mortality in the affected individuals. Melanoma of different tissue origins, such as the uvea, conjunctiva, eyelid, orbit, and lacrimal sac, is distinct in terms of patient characteristics, manifestations, and disease prognosis. The treatment modality varies, based on the location and extent of the lesion as well as the stage. Despite prompt diagnosis and management, local recurrence and metastasis occur in a number of patients. In general, melanoma predominantly affects the Caucasian population. Thereby, there is a lack of overall investigations on ocular melanoma in the Asian population. In this study, we analyze the epidemiology, clinical manifestation, treatment strategy, and long-term outcome of ocular melanoma in a tertiary referral center in Taiwan.

## 2. Materials and Methods

### 2.1. Patient Enrollment

This retrospective series enrolled patients with a clinical and confirmatory pathological diagnosis of ocular melanoma who were managed at Taipei Veterans General Hospital between 1 January 1985 and 31 August 2019. The study was conducted in accordance with the Declaration of Helsinki and was approved by the local ethics committee of the institute.

### 2.2. Data Collection

A thorough review of the medical records was conducted. Demographic data included patient age and gender. The tumor origin was identified. Management consisted of local treatment for the primary tumor with/without adjuvant therapy. Outcomes were represented by disease local recurrence, distant metastasis, melanoma-specific survival, as well as melanoma-related mortality throughout the follow-up period. Additionally, the intervals between diagnosis and metastasis as well as between metastasis and mortality were displayed.

### 2.3. Statistical Analysis

SPSS computer statistical software (version 20.0; SPSS, Chicago, IL, USA) was used for statistical analysis. Group differences were analyzed using the two-tailed Fisher’s exact test and the Mann–Whitney test. The Kaplan–Meier method was used to analyze the local recurrence, metastasis, and mortality rate. A *p*-value < 0.05 was considered statistically significant.

## 3. Results

A total of 88 consecutive patients diagnosed with ocular melanoma were included in this study. The median follow-up time was 41.0 months. The demographics of the study population are summarized in Table 1. Melanoma arose most commonly from the uvea (*n* = 55, 62.5%), followed by the conjunctiva (*n* = 18, 20.5%), orbit (*n* = 8, 9.1%), eyelid (*n* = 6, 6.8%), and lacrimal sac (*n* = 1, 1.1%). There were 47 (53%) men and 41 (47%) women. Patients with eyelid, conjunctival, or orbital melanoma were predominantly male, while more female patients were demonstrated in uveal melanoma. The median age at presentation was 54.5 years. Patients with uveal melanoma were younger (median, 47.0 years) than those with melanoma located elsewhere, including the conjunctiva (median, 66.0 years), eyelid (median, 78.5 years), and orbit (median age, 79.5 years).

The treatment modality and clinical outcome are illustrated in Table 2. Uveal melanoma was primarily treated with enucleation (*n* = 44, 80%). Other treatment modalities consisted of exenteration (3.6%), gamma knife (7.3%), and transpupillary thermal therapy (7.3%). Conjunctival melanoma was treated with either exenteration (55.6%) or wide excision (38.9%). For eyelid, orbital, or lacrimal sac melanoma, wide excision was first considered. Adjunctive treatment with chemotherapy and/or radiotherapy was conducted in 34% of the study population.

Overall, local recurrent diseases occurred in 9% of patients. The local recurrence rate was higher for melanoma of the orbit (38%) and eyelid (33%) than for that of the uvea (4%) and conjunctiva (6%). Local recurrence was treated with complete resection, radiotherapy, exenteration, enucleation, or a combination of these modalities. A total of 38 patients (43%) developed metastasis during a median follow-up period of 41.0 months (IQR, 19–77 months). Throughout this period, melanoma-associated death occurred in 36 patients (41%); mortality from lacrimal sac and orbital melanomas was the highest (100% and 50%, respectively).

Overall, 38 patients (43%) developed metastasis (Table 3). Seven patients (18%) were diagnosed with a metastatic disease at presentation. Among them, five had uveal melanoma and two had conjunctival melanoma. The other 31 patients developed metastasis in a median duration of 27.2 months (IQR, 13–58 months) from diagnosis. The time from metastasis to mortality was generally short, with 7.0 months on average (IQR, 3–15 months), except for eyelid melanoma (median, 74.0 months), which had a longer duration from diagnosis to metastasis. Patients died in a median of 34.0 months (IQR, 13–59 months) after diagnosis. Metastatic disease occurred most commonly in the liver (65.8%), followed by bone (31.6%), lung (26.3%), lymph nodes (18.4%), brain (7.9%), and skin (2.6%).

The distribution of the gender and age of patients with melanoma originating from a specific location is illustrated in Appendix A. Overall, male patients presented at an older age compared to female patients (*p* < 0.01).

By Kaplan–Meier survival estimates, the melanoma-specific survival rate of ocular melanoma at 1, 3, 5, and 10 years was 89.8%, 73.3%, 58.4%, and 43.7%, respectively (Figure 1, Table 4).

## 4. Discussion

In our study conducted in a single medical center, ocular melanoma occurred most frequently in the uvea (62.5%) and conjunctiva (20.5%), although melanoma arising from the orbit (9.1%), eyelid (6.8%), and lacrimal sac (1.1%) was possible. An analysis from the United States illustrated that among ocular melanomas, 85.0% were uveal, 4.8% were conjunctival, and 10.2% occurred at other sites [1,3,4,5]. There was found to be a difference in the distribution of the primary location of ocular melanoma between Asians and Caucasians. An analysis in the Asian cohort from Singapore reported that uveal and conjunctival melanomas accounted for 65.9% and 34.1%, respectively [6]. This was comparable with the finding in our cohort that uveal melanoma comprised 62.5% of all ocular melanoma, suggesting a higher proportion of melanoma originating from sites other than the uvea compared with that from the uvea in Asians.

In our cohort, the general age at diagnosis of ocular melanoma was 54.5 years on average, wherein patients with uveal melanoma were diagnosed at a younger age (median, 47.0 years) than those with melanoma of other origins. The mean presenting age of ocular melanoma in the Asian cohort from Singapore, however, was 50 years [6]. As compared to the Asian patients, the reported age at presentation peaked in the seventh and eighth decade of life in Caucasians [3,7]. With regard to melanoma of a specific location, the presenting age differs among different ethnic groups as well. Uveal melanoma was diagnosed at approximately 60 years of age in the United States and Britain [8,9], whilst the age of diagnosis ranged from 41 to 53 years in Asian populations [6,10]. Conjunctival melanoma has persistently been diagnosed at younger ages in Asians compared to Caucasians. In our study, the median age of the diagnosis of conjunctival melanoma was 66.0 years. In the literature, the presenting age ranged from 55 to 65 years in Caucasians and from 43 to 54 years in Asians [6,11,12]. In our study, the median age of eyelid melanoma was 78.5 years, while Caucasian patients have been reported to be diagnosed at 65 years [13]. With respect to orbital melanoma, the reported age varies considerably from 5 to 87 years [14]. In our cohort, orbital melanoma presented at 79.5 years on average. There was discrepancy in the age of diagnosis of ocular melanoma between men and women as well. Male patients were older (60.6 years) compared with female patients (50.4 years) (Appendix A). In contrast, female patients tended to present at an older age in the United States [3].

Except for uveal melanoma, which showed female predominance, we demonstrated a male predominance in conjunctival (61%), eyelid (67%), and orbital (75%) melanomas. This was consistent with the literature, where the incidence of ocular melanoma was higher in men than in women and the male to female rate ratio was 1.29 [8]. A male predilection was also presented in conjunctival, eyelid, and orbital melanomas [15]. Studies on Chinese and Taiwanese populations with conjunctival melanoma reported that men comprised 74% and 60% of patients, respectively [12,16]. For eyelid melanoma, males comprised 53% of all subjects [15]. The gender difference of conjunctival and eyelid melanomas was believed to be attributed to the excessive ultraviolet exposure of men [15]. In accordance with the review on primary orbital melanoma, males were affected more often than females, although the difference was less obvious in the review article (male 58%, female 42%) as compared to that in the current study (male 75%, female 25%). Men also showed a higher incidence of uveal melanoma than women in the United States [16]. In studies of the Asian population, the prevalence of male patients was variable, ranging from 40% to 61% [6,10]. In our study, men comprised 47% of patients with uveal melanoma.

Management of melanoma depends on the location, size, and extent of the tumor as well as the presence of metastasis, therapeutic modalities, the surgeon’s expertise, and the patient’s concerns [8]. Enucleation is the definitive treatment for primary intraocular tumors, whilst eye-sparing modalities, e.g., brachytherapy and proton beam radiotherapy, have been increasingly introduced [8]. Wide excision with or without adjuvant therapy is primarily adopted for conjunctival and eyelid melanomas. In our study, the majority of subjects with uveal melanoma were treated with enucleation (80%), while more than half of the populations with conjunctival melanoma required exenteration (56%). Also, exenteration or excision/debulking was considered the major treatment modality for orbital melanoma [14]. Because of the paucity of evidence on orbital and lacrimal sac melanomas, no standard treatment has been established, though local resection or exenteration is commonly employed [14].

In the current study, the overall local recurrence rate was 9% at a median follow-up of 41.0 months. The cumulative local recurrence of uveal melanoma (4%) was lower than that of eyelid melanoma (33%) at 5 years, while 6% of patients with conjunctival melanoma and 38% of patients with orbital melanoma experienced recurrences at 2 years and 2.9 years, respectively. The relatively lower local recurrence for uveal and conjunctival melanomas in our study may be associated in part with a greater portion of patients who underwent enucleation and exenteration, respectively. Cerman et al. reported that among patients with choroidal melanoma following photodynamic therapy, 21% developed local recurrence over 31 months [17], while 9–28% of choroidal melanoma recurred after transpupillary thermotherapy [18,19]. One multicenter study recruiting data from ten ophthalmic oncology centers from nine countries reported that the local recurrence of conjunctival melanoma after local resection was 5.4%, 19.3%, and 36.9% at 1, 5, and 10 years, respectively [20]. A higher local recurrence rate has been reported in the Chinese population, wherein the 1-, 5-, and 10-year recurrence was 31.0%, 59.7%, and 66.4%, respectively [12]. In one retrospective cohort study in a medical center in Taiwan, 30% of patients with conjunctival melanoma developed local recurrence in five years [16]. Orbital and eyelid melanomas recurred most frequently (38% and 33%, respectively). Chan et al. reported that 17% of eyelid melanoma recurred locally throughout a median duration of three years after tumor excision [13], which may result from incomplete excision [21]. One review on 88 patients of primary orbital melanoma reported that 15% of them developed local recurrence throughout an average of three years after treatment [14].

The metastasis rate was generally high. Forty-three percent of all the patients with ocular melanoma developed metastasis in an averaged duration of 27.2 months (Table 3). The only patient with lacrimal sac melanoma developed metastasis at 17 months after diagnosis. The metastasis rate was 50% for orbital melanoma, followed by uveal, conjunctival, and eyelid melanomas (44%, 39%, and 33%, respectively). Meanwhile, metastasis occurred most rapidly after diagnosis in patients with orbital melanoma (6 months). The duration was relatively long for eyelid melanoma (74 months). The reported metastasis rate varies across studies due to distinct circumstances. The overall metastasis of uveal and conjunctival melanoma was 38.5% in the median 154-month period in an Asian population [6]. With regard to conjunctival melanoma, it has been reported to be 16% at five years and 26% at ten years in Caucasians [22]. Investigations of the Chinese population showed that the 1-, 5-, and 10-year metastasis rate was 16.7%, 38.7%, and 50.9%, respectively [12], while 60% of Taiwanese patients with conjunctival melanoma exhibited distant metastasis in 59.5 months [16]. In accordance with previous results, the metastasis rate was higher in Asians than in Caucasians. A review on 179 cases of periocular cutaneous melanoma indicated 12% metastasis through a mean of 55.7 months [23]. The metastasis rate of primary orbital melanoma was measured at 36% in 3 years [14]. Relatively fewer metastasis were demonstrated in eyelid, conjunctiva, and uveal melanomas (33%, 39%, and 44%, respectively) as compared with lacrimal sac and orbital melanomas (100% and 50%, respectively). This finding may be attributed in part to early recognition and prompt management. Uveal melanoma tends to affect vision. Common manifestations include blurred vision, floaters, and visual field defects [6]. On the other hand, conjunctival and eyelid melanomas may be apparent, particularly when they are hyperpigmented and/or located at sites that are easily detected. A delay in identification of tumors located in the lacrimal sac or orbit is highly possible because of the vague manifestation early on. Evidence indicates that orbital melanoma typically presents with painless proptosis, which is hard to discover until obvious growth [14]. Although gross metastasis was not detected in patients with lacrimal sac or orbital melanoma at presentation, there may be micrometastasis. Additionally, eyelid melanoma patients developed metastasis at a median duration of 74 months after presentation. Therefore, long-term surveillance is necessary.

In the current study, the overall mortality throughout a 41-month follow-up was 41% (Table 2). Forty percent of patients with uveal melanoma died over 46.5 months. The mortality rate was variable among cohorts, ranging from 5% to 45% for uveal melanoma [24]. In an Asian cohort, 23.1% of patients died at a median duration of 154 months [6]. Although the survival rate of uveal melanoma has been suggested to be higher in Asians than in Caucasians, our study demonstrated worse survival outcomes in the Taiwanese population [6]. Conjunctival melanoma-related death after surgical resection was 18% over a mean interval of 4.9 years in the Caucasian population [25], whereas in the Chinese population, the 1-, 5-, and 10-year mortality was 3.8%, 30.5%, and 37.4%, respectively [12]. In our study, 39% of patients with conjunctival melanoma died in a median follow-up of 26.0 months. This was comparable to the result from one Taiwanese cohort, wherein 50% of patients died in a mean follow-up time of 68.7 months [16]. Corresponding to previous findings, there were poorer survival outcomes in Asians compared to Caucasians, which is believed to be related to the advanced stages at presentation and unawareness of conjunctival melanoma in Asians [6]. Death from eyelid melanoma was reported at 4.9% in a mean duration of 55.7 months in Caucasians and 9.8% in a median duration of 127 months in Chinese [23]. In our study, the mortality of eyelid melanoma was 33% in a median 76-month follow-up. Mortality was the highest for lacrimal sac and orbital melanomas. The only one patient with lacrimal sac melanoma died 35 months after diagnosis, while four out of eight patients (50%) with orbital melanoma died in 18 months. In the literature, the averaged mortality rate was 38% over 4.5 years [14]. The high mortality rate of lacrimal sac and orbital melanomas may be associated with the concurrent high local recurrence rate of these malignancies.

Treatment for metastatic melanomas is a challenge for all clinicians. Recent studies have shown a significantly higher success rate with a combination of immunotherapy and chemotherapy, or targeted therapy [26]. Immunotherapy such as ipilimumab (a monoclonal antibody to cytotoxic T-lymphocyte antigen 4, CTLA-4), anti-PD1 (programmed cell death protein 1), or combining both, has been shown to have a beneficial effect on survival in the treatment of metastatic skin melanoma; however, it remains to be determined whether this also affects ocular melanomas [27]. Targeted therapy that interferes with a specific molecular pathway to inhibit tumor angiogenesis, invasion, and metastasis is another promising treatment for advanced or metastatic ocular melanomas [27,28,29].

Primary melanoma of the lacrimal sac is extremely rare; the incidence accounts for 0.7% of all ocular melanomas [30], and only one patient with this melanoma was reported in our cohort, a 57-year-old woman presenting with tearing in both eyes for 3 months. Nasolacrimal duct obstruction was initially impressed and underwent balloon dacryoplasty, but in vain. She then developed bloody tearing and local swelling over the left lacrimal sac region and came to our clinic. Orbital computed tomography revealed a 14 mm ovoid shape soft tissue mass in the left lacrimal sac with the involvement of the nasolacrimal duct (Figure 2). A tentative biopsy revealed a black-pigmented mass that was pathologically confirmed to be malignant melanoma (Figure 3A). A wide excision of the lacrimal sac tumor and anterior ethmoidectomy and medial maxillectomy was performed (Figure 3B). Despite adjuvant chemotherapy with radiotherapy, the patient developed lung and multiple bone metastasis and died at 18 months after metastasis. Lacrimal sac melanoma presents insidiously and masquerade as chronic dacryocystitis, with early symptoms such as epiphora leading to misdiagnosis. Tearing with blood reflux and a mass above the medial canthal tendon should alert clinicians of lacrimal sac malignancy [31]. The low index of suspicion and diagnostic delay contribute to the spread of the tumor through vascular and nasolacrimal routes with adjacent orbital structures with poor survival outcomes [32,33].

Almost all patients developing metastasis died (Table 3), which may be attributed to the current limitations of effective systemic therapy for metastasis disease. Diener-West et al. reported that death from metastasis of uveal melanoma was 80% at one year and 92% at two years [34]. In our study, the duration between metastasis and death was 7.0 months on average. The duration was shorter for orbital and eyelid melanomas (4.0 months and 6.5 months, respectively) compared with melanoma from the uvea, conjunctiva, or lacrimal sac. We believe that the older age of patients with melanoma of these two origins was responsible in part for the rapid mortality after metastasis.

## 5. Conclusions

Ocular melanoma arose most commonly from the uvea, followed by the conjunctiva, orbit, eyelid, and lacrimal sac. There was a male predominance except for uveal melanoma. The overall presenting age was 54.5 years. Patients with uveal melanoma were younger than those with ocular melanoma located elsewhere. The local recurrence rate was 9% during the 41-month follow-up period. Local recurrence occurred more commonly in patients with orbital or eyelid melanoma than in those with uveal or conjunctival melanoma. Forty-three percent of the subjects developed metastasis, which occurred in a median duration of 27.2 months from diagnosis. In addition, most of the patients with metastatic disease died, while the overall mortality was 41%. This study provided the epidemiology and treatment outcome of ocular melanoma. Early diagnosis and effective management were warranted.

## Figures and Tables

**Figure 1 cancers-13-05926-f001:**
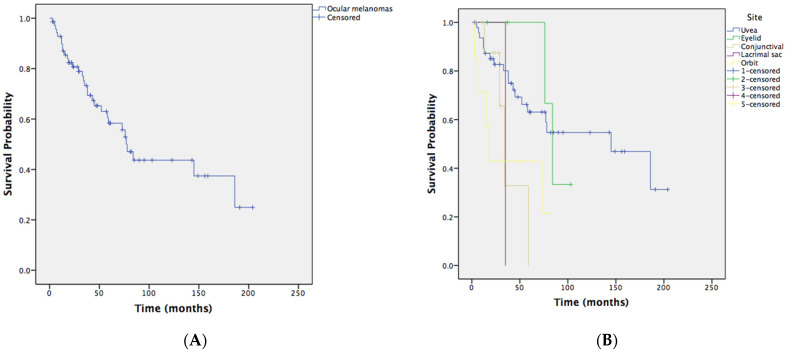
Kaplan–Meier estimate of melanoma-specific survival in patients with primary ocular melanoma. (**A**) Melanoma-specific survival probability. (**B**) Melanoma-specific survival probability in the subgroups with melanoma of specific origins.

**Figure 2 cancers-13-05926-f002:**
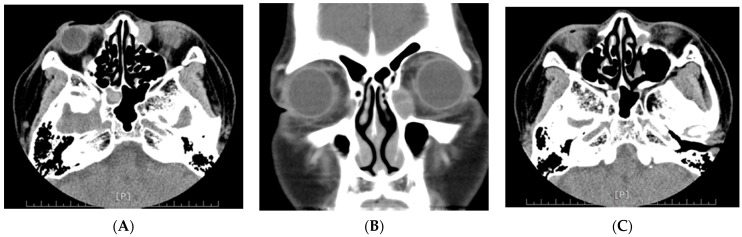
Orbital computed tomography of lacrimal sac melanoma. (**A**,**B**) A 14 mm ovoid shape soft tissue mass in left lacrimal fossa. (**C**) Lacrimal sac tumor with involvement of nasolacrimal duct.

**Figure 3 cancers-13-05926-f003:**
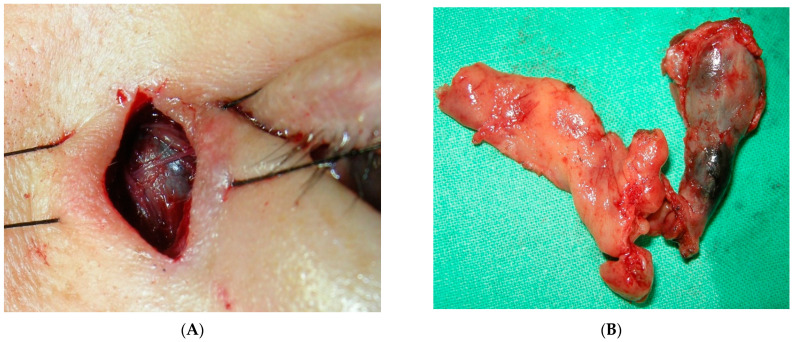
Lacrimal sac melanoma. (**A**) One black-pigmented mass noted over left lacrimal sac. (**B**) Wide excision of lacrimal sac and nasolacrimal duct tumor with adjacent soft tissue.

**Table 1 cancers-13-05926-t001:** Demographics of subjects with ocular melanoma.

Tumor Origin	*n* (%)	Gender, *n* (M/F)	Median Age, Years (IQR)	Laterality, *n* (R/L/B)
Uvea	55 (62.5%)	26/29	47.0 (42–60)	30/25/0
Eyelid	6 (6.8%)	4/2	78.5 (56–82)	3/3/0
Conjunctiva	18 (20.5%)	11/7	66.0 (46–75)	11/7/0
Lacrimal sac	1 (1.1%)	0/1	57.0 (NA)	0/1/0
Orbit	8 (9.1%)	6/2	79.5 (77–85)	4/3/1
Total	88 (100%)	47/41	54.5 (44–72)	48/39/1

B: Both sides, F: female, L: left, M: male, R: right, IQR: interquartile range. NA: not applicable.

**Table 2 cancers-13-05926-t002:** Treatment modalities and outcomes of ocular melanoma.

Tumor Origin	Treatment				Local Recurrence	Metastasis	Mortality	Follow-Up
	Wide Excision	Exenteration/Enucleation	Chemotherapy/Radiation/CCRT	Gamma Knife/TTT	*n* (%)	*n* (%)	*n* (%)	Months Median (IQR)
Uvea	0	2/44	13/3/0	4/4	2 (4%)	24 (44%)	22 (40%)	46.5 (22–84)
Eyelid	3	1/0	1/0/0	0/0	2 (33%)	2 (33%)	2 (33%)	76.0 (27–94)
Conjunctiva	7	10/0	1/0/1	0/0	1 (6%)	7 (39%)	7 (39%)	26.0 (13–31)
Lacrimal sac	1	0/0	0/0/1	0/0	0 (0%)	1 (100%)	1 (100%)	35.0 (NA)
Orbit	5	2/0	0/4/2	0/0	3 (38%)	4 (50%)	4 (50%)	18 (6–73)
Total	16	15/44	15/7/4	4/4	8 (9%)	38 (43%)	36 (41%)	41.0 (19–77)

CCRT: Concurrent chemoradiotherapy, TTT: transpupillary thermotherapy, IQR: interquartile range, NA: not applicable.

**Table 3 cancers-13-05926-t003:** Metastasis, mortality, and the time interval.

Tumor Origin	Metastasis, *n* (%)	Metastasis at Dx, *n*	Dx to Metastasis, Months *	Metastasis to Mortality, Months	Dx to Mortality, Months
Uvea	24 (44%)	5	30.5	7.0	38.0
Eyelid	2 (33%)	0	74.0	6.5	80.0
Conjunctiva	7 (39%)	2	16.1	9.0	31.5
Lacrimal sac	1 (100%)	0	17.0	18.0	35.0
Orbit	4 (50%)	0	6.0	4.0	10.5
Total	38 (43%)	7	27.2	7.0	34.0

Dx: Diagnosis. *: Dx to metastasis among 31 patients (excluded 7 metastasis noted at presentation).

**Table 4 cancers-13-05926-t004:** The 1-, 3-, 5-, and 10-year melanoma-specific survival of primary ocular melanoma.

Tumor Origin	1-Year Survival Rate	3-Year Survival Rate	5-Year Survival Rate	10-Year Survival Rate
Uvea	89.4%	80.1%	63.1%	54.7%
Eyelid	100%	100%	100%	33.3%
Conjunctiva	100%	32.8%	0%	0%
Lacrimal sac	100%	0%	0%	0%
Orbit	71.4%	42.9%	42.9%	21.4%
Total	89.8%	73.3%	58.4%	43.7%

## Data Availability

The data that support the findings of this study are available from the corresponding author upon reasonable request.

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
