# Peer review of "The Clinical Feature and Treatment Outcome of Ocular Melanoma: A 34-Year Experience in a Tertiary Referral Center"

_cancers, 2021, doi:10.3390/cancers13235926_

Round 1

Reviewer 1 Report

This manuscript reports the results of the review of the clinical data of ocular melanoma patients referring to a single hospital in Taipei. The study is clear and well presented but no genetic data on the tumors are reported. On my opinion the part related to the treatments at relapse can be better detailed.

Author Response

We appreciate your comments and we modify our manuscript with adding details on discussion of treatment for relapse of melanoma on discussion part (line 263-271).

“Treatment for metastatic melanomas is a challenge for all clinicians. Recent studies have shown a significantly higher success rate with combination of immunotherapy and chemotherapy, or targeted therapy. Immunotherapy such as Ipilimumab (a monoclonal antibody to cytotoxic T-lymphocyte antigen 4, CTLA-4), Anti-PD1 (programmed cell death protein 1), or combining both, has been shown to have a beneficial effect on the survival in the treatment of metastatic skin melanoma; however, it remains to be determined whether this also effects on ocular melanomas. Targeted therapy that interferes with a specific molecular pathway to inhibit tumor angiogenesis, invasion and metastasis, is another promising treatment for advanced or metastatic ocular melanomas.”

Reviewer 2 Report

It was with pleasure I read the manuscript by Dr Huang et al concerning ocular melanoma in an Asian population.  My comments are:

  1. Change ”Recurrence” to ”Local recurrence” through-out the manuscript, it now becomes confusing if it refers to recurrence in general or local recurrence. E.g. in the conclusion it is stated that ” The recurrence rate was 9% 268 during the 55.4-months follow-up period”, but that is only for local recurrence and not for metastatis disease.
  2. Change ”cases” to ”patients throughout the manuscript.
  3. In Methods it has to be defined what is defined as mortality, eg. in Table 2 there is 4 patients with metastasis from orbit melanoma, but 5 patients that died. Did they die from other causes than melanoma? I would recommend to be stringent and define ”Local recurrence”, ”Distant metastasis”, ”Melanoma-specific survival” and ”Overall survival” to match with the most commonly reported outcomes.
  4. Table 4 do not add very much information, consider moving to supplementary.

Author Response

We appreciate your comments and some modifications were applied to our manuscripts as below.

  1. We changed “recurrence” to “local recurrence” throughout the manuscript and abstract.
  2. We changed “cases” to “patients” throughout our manuscript.
  3. We refined the definition in methods. “Outcomes were represented by disease local recurrence, distant metastasis, melanoma-specific survival, as well as melanoma-related mortality throughout the follow-up period.” (line 64-66).

To exclude one patient of mortality that died from other causes other than melanoma, we correct the melanoma-related mortality of metastasis of orbital melanoma to be 4 patients (50%) as well as melanoma-related mortality to be 36 patients (41%) throughout the manuscript and table as well.

  1. We moved Table 4 to supplementary.

Reviewer 3 Report

This is an interesting manuscript reporting on the experience of a tertiary referral center on ocular melanoma.

I think in the text it should be stated more clearly, especially in the discussion, that there was only 1 case of lacrimal sac melanoma included in the study.

Table 3: Was the time from diagnosis to occurence of metastasis only calculated out of the patients that had no metastases at the time point of diagnosis (should be 31) or are the values provided the means for all 38 patients presenting with metastases at any time point? Please clarify.

Author Response

We appreciate your comments and some modifications were applied to our manuscript as below.

  1. We add detailed information about the patient of lacrimal sac melanoma with figures and discuss the challenge in the diagnosis and management of lacrimal sac melanoma. (line 272-308)
  2. The time from diagnosis to metastasis accounted only 31 patients (except metastasis at diagnosis). We rewrote the draft and make it clear. “Overall, 38 patients (43%) developed metastasis (Table 3). Seven patients (18%) were diagnosed with a metastatic disease at presentation. Among them, five had uveal melanoma and two had conjunctival melanoma. Other 31 patients developed metastasis in a mean duration of 27.2 months (range, 2–178 months) from diagnosis.” (line 106-110) We also added an explanation on Table 3. “Dx to Metastasis, months*” and ”*: Dx to Metastasis among 31 patients (excluded 7 metastasis noted at presentation).”

Round 2

Reviewer 1 Report

The authors have improved the manuscript.

Author Response

Thanks for your comment

Reviewer 2 Report

Dear authors,

my comments have been adressed fully. Unfortunately, and I do apologize for this, I noted now at a second read-through that age and times (follow-up, survival etc) are presented as mean values with ranges. Since all of these data are non-parametric, the correct way from a statistical point-of-view would be to present the results as median and interquartile range.

Author Response

We appreciate your comments and we refined the statistical results of age and follow-up time as median and interquartile range throughout the content, table, and abstract as well.